# Tannin-Derived Hard Carbon for Stable Lithium-Ion Anode

**DOI:** 10.3390/molecules27206994

**Published:** 2022-10-18

**Authors:** Ming-Jun He, Lai-Qiang Xu, Bing Feng, Jin-Bo Hu, Shan-Shan Chang, Gong-Gang Liu, Yuan Liu, Bing-Hui Xu

**Affiliations:** 1College of Materials Science and Engineering, Central South University of Forestry and Technology, Changsha 410004, China; 23rd Division Convergence Media Center, Tumushuke 843900, China; 3Center Astrum Innovations Limited, Wisdom Park, Country Garden, Changsha 410006, China; 4Institute of Materials for Energy and Environment, School of Materials Science and Engineering, Qingdao University, Qingdao 266071, China

**Keywords:** hard carbon, tannin, lithium-ion battery, graphene oxide

## Abstract

Graphite anodes are well established for commercial use in lithium-ion battery systems. However, the limited capacity of graphite limits the further development of lithium-ion batteries. Hard carbon obtained from biomass is a highly promising anode material, with the advantage of enriched microcrystalline structure characteristics for better lithium storage. Tannin, a secondary product of metabolism during plant growth, has a rich source on earth. But the mechanism of hard carbon obtained from its derivation in lithium-ion batteries has been little studied. This paper successfully applied the hard carbon obtained from tannin as anode and illustrated the relationship between its structure and lithium storage performance. Meanwhile, to further enhance the performance, graphene oxide is skillfully compounded. The contact with the electrolyte and the charge transfer capability are effectively enhanced, then the capacity of PVP-HC is 255.5 mAh g^−1^ after 200 cycles at a current density of 400 mA g^−1^, with a capacity retention rate of 91.25%. The present work lays the foundation and opens up ideas for the application of biomass-derived hard carbon in lithium anodes.

## 1. Introduction

Traditional energy consumption has brought about many unexpected problems such as environmental pollution. Developing clean energy and gradually realizing the transformation of energy structure is the development goal of the new era [1,2]. The development of large-scale energy storage and powered vehicles has greatly stimulated the development of lithium-ion batteries (LIBs) [3,4,5,6]. Up to now, lithium-ion batteries have been flourishing as one of the most important energy storage methods nowadays [7,8]. Among the anode materials for LIBs, the most mature one is graphite [9,10,11,12], which has good performance in all aspects. However, in the long-term development, it is proved that the mining and utilization of graphite can cause environmental pollution. The synthesis of graphite usually requires high temperature, which leads to the emission of various gases, such as nitrogen oxides and carbon dioxide. Therefore, different alternative anode materials have been sought.

A large number of experiments have proven that hard carbon is one of the most promising materials for LIBs [8,13,14]. Hard carbon, as a non-graphitized carbon material, has a large number of disordered graphite layers in its structure, and the increased layer spacing allows effective lithium storage [15,16]. Many materials are used for the synthesis of hard carbon, e.g., waste engineering plastics [17], biomass [18,19,20], phenolic resins [21]. Hard carbon obtained based on renewable biomass is a very promising direction, with the advantages of abundant source and environmental protection [14,22].There are still many problems with biomass as a precursor of hard carbon for LIBs: 1. The naturally occurring biomass in nature is not fully utilized every year; 2. The mechanism of lithium storage in hard carbon formed from biomass is still unclear.

Tannin is a secondary organism produced by plant metabolism during plant growth. They are widely found in the bark or fruit of plants and are easily extracted to obtain the product [23,24]. As a rich source of biomass, hard carbon derived from tannin is still little studied in lithium-ion battery anode materials. The suitability of tannin as a hard carbon precursor is worthy of in-depth study. Based on this, this thesis investigates the performance of hard carbon in lithium-ion batteries obtained by cleavage of tannin in argon gas. Subsequently, further by compounding with graphene oxide (GO), GO/hard carbon composites were obtained to take advantage of the advantages of both and effectively improve the performance. The results demonstrate that the composite in the PVP system behaved better performance. We summarized the lithium storage mechanism inside this system, which provides the basis for the later work.

## 2. Results and Discussion

As depicted in Figure 1a, the whole process of synthesizing hard carbon is simple. Tannin is extracted from rich tree sources and calcined to obtain hard carbon for lithium-ion anodes. To investigate the change process of tannin getting hard carbon in an argon environment, the thermogravimetric test was performed in-depth. From the thermogravimetric results, it can be seen that the curves of tannin extracts obtained from acacia equisetum bark were similar to the decomposition curves of tannin obtained from extracts of other plants [25]. At the same time, the process of tannin decomposition is divided into several stages (Figure 1b). From room temperature to ~180 °C, it is attributed to the release of the volatility of volatile components (e.g., H_2_O). Secondly, in the range of ~180 °C to ~400 °C, it is caused by the rapid decomposition of precursor. After ~400 °C, the process was belonged to the intermolecular crosslinking, which promotes carbon formation. A large amount of gas (CO_2_, H_2_O) is produced at this stage [26].Finally, as the temperature continues to rise, the mass is no longer changed, implying the formation of hard carbon. It can be concluded that the carbon yield is about 48.9%. The carbon yield is associated with the properties of biomass. The carbon yield reported in this research is apparently higher than other biomass: sucrose [27], cellulose. This advantage is conducive to the macro preparation of hard carbon.

To probe the structural properties, intrinsic nature of tannin-derived hard carbon, XRD was firstly carried out, as shown in Figure 1c. The peaks at ~23.8 and ~44.8 degrees are assigned to the (002) and (100) planes of graphite [28,29], respectively. The d_002_ distance of the hard carbon is calculated by the Bragg equations, it can be 0.373 nm. Compared to the d_002_ distance of graphite (0.335 nm), the increased layer spacing facilitates the intercalation of lithium ions and enhances the lithium storage performance of the material. Except for the peaks of (002) and (100) planes, other peaks marked in the XRD patterns are belonged to the impurities, which are associated with the nature of biomass. According to results reported by other researchers, the impurities are mostly assigned to the inorganic substances belonging to the combination of CaS, K_2_CO_3_ [25].

To investigate the porosity of the obtained hard carbon, BET testing was conducted which is shown in Figure 1d,e. The specific surface area of the HC is 0.30 m^2^/g. The very low specific surface area of hard carbon can effectively reduce side reactions with the electrolyte and is expected to achieve high first coulombic efficiency [30,31,32]. Moreover, as shown in Figure 1e, the pore size distribution indicates that the obtained hard carbon is almost free of micropores and mesopores, corresponding to the low specific surface area.

To directly observe the morphology of hard carbon, scanning electron microscopy tests were performed which are displayed in Figure 2. The hard carbon shows a random morphology, in which the size of the particles is in the micron range (Figure 2a–c). At the same time, the surface of the particles is smooth with no obvious pores, which is consistent with the previous BET results. Furthermore, the morphologies of two graphene oxide/hard carbon composites obtained by the previous different treatments are showed similar trend without any major changes in the morphology of the hard carbon itself (Figure 2c–h). Part of the graphene oxide was successfully wrapped on the hard carbon surface, reflected in the enhancement of the electrochemical properties later. As depicted in Appendix A, the composite hard carbon has similar XRD curves with HC, indicating that the structure of hard carbon is maintained. The (002) peaks of CTAB-HC and PVP-HC are shifted to a lower degree (~23.6), which facilitates ion insertion. BET testing was also conducted to confirm the influence caused by the introduction of graphene oxide. The specific surface area of composite materials is both improved compared to the HC, in which CTAB-HC and PVP-HC can reach 0.9425 and 3.0676 m^2^/g, respectively (Appendix A). The introduction of graphene oxide is helpful to create more mesopore for the system. Specially, this is more evident in the PVP-HC system (Appendix A). The increase in specific surface area facilitates the infiltration of electrode solution and accelerates ion transport [33,34].

The TEM images state that the hard carbon shows a non-graphitic and turbostratic structure (Appendix A). The special structure is related to the (002) crystalline surface spacing, turbostratic domains can effectively increase the crystal spacing. This trend can be demonstrated by the XRD results. Furthermore, through the inverse Fourier transform (Appendix A), the d_002_ is calculated to be 0.382 nm. The larger d_002_ than the graphite is advantageous to facilitate the entry of lithium ions into the interlayer [35,36]. From the TEM images (Figure 3a), it is shown that CTAB-HC is consist of large spherical particles, in agreement with SEM images. Towards a high resolution, it is observed that the surface of HC is wrapped with GO (Figure 3b–c). HRTEM furthermore demonstrated the composite structure of CTAB-HC. The red mark in Figure 3d is corresponding to the existence of GO, in which the interlayer spacing of 0.34 nm is belonged to GO (Figure 3d). At the same time, the TEM image of PVP-HC also displayed the morphology of spherical (Figure 3e). Compared with CTAB-HC, GO is more uniformly distributed on the surface of PVP-HC, contributing to superior electrochemical performance (Figure 3f,g). For the HRTEM image of PVP-HC, it is clearly seen that the structure is composed of GO and hard carbon, depicted in Figure 3h. All SAED patterns show diffuse diffraction rings, which further indicate the structure of turbostratic (Appendix A).

The electrochemical properties of tannin-based hard carbon were evaluated using coin cells with lithium metal as a counter electrode. The specific electrochemical redox process of hard carbon is described in the cyclic voltammetry curve, as exhibited in Appendix A. The reduction peaks at the first cycle (~0.6 V, ~1.29 V, ~1.71 V) are associated with the formation of solid electrolyte interphase [37,38], which is irreversible process. Cyclic voltammetry curves of PVP-HC (Figure 4a) and CTAB-HC (Appendix A) are similar to HC. Initial coulombic efficiency is related to the process, affecting the reversible capacity. The reaction process curves for the second and third laps are very similar, representing that the subsequent charging and discharging process is relatively stable. Long-time cycling was performed to test the specific electrochemical properties (Figure 4b). Without the introduction of GO, the pure hard carbon exhibits a capacity of 305.8 mAh g^−1^ (current density = 400 mA g^−1^) at the first discharge process. The reversible charge capacity is 255.9 mAh g^−1^. After 200 cycles, the capacity can reach 218.1 mAh g^−1^, the capacity retention is 85.2%. TEM of the cycled cell revealed no significant changes in the overall structure (Appendix A).

To further improve the capacity of the hard carbon, constructing composite electrode materials with GO was performed as a method. The capacity of PVP-HC is apparently enhanced compared to the pure HC. At the current density of 400 mA g^−1^, the reversible capacity of PVP-HC is 255.5 mAh g^−1^ after 200 cycles, which is 17.14% higher than the pure HC. Within the CTAB-HC system, GO monolithic cladding of hard carbon is less effective, resulting in a less-than-obvious lifting effect. The cycling performance improvement comes from the addition of GO to enhance the electrolyte wetting effect and ion kinetics improvement. Meanwhile, the compounding of GO can bring about a decrease in the initial coulombic efficiency. Due to the low porosity of HC, the initial coulombic efficiency is 83.68%. The specific surface area of the composite materials is improved, leading to that the initial coulombic efficiency of CTAB-HC and PVP-HC decreasing to 81.95%, 70.59% (Figure 4c). Then rate performance is also compared as shown in Figure 4d. The capacity of HC is 322.74, 249.08, 189.81, 146.12, 130.54 mAh g^−1^ at the current density of 100, 200, 400, 800, 1000, 2000 mA g^−1^. The capacity of HC is 306.76 when the current density returns to 100 mAh g^−1^. Compared to the pure HC, the rate performance of PVP-HC shows an obvious enhancement, in which the capacity is 337.92, 270.36, 234.72, 202.68, 191.88, 156.12 mAh g^−1^ at the current density of 100, 200, 400, 800, 1000, 2000 mA g^−1^.

To investigate the intrinsic mechanism between materials and the electrochemical performance, the characteristics of the charging and discharging process are analyzed. Clearly, the charge/discharge curves are similar (Figure 5a). The capacity contributed by different voltage intervals is calculated precisely, in which PVP-HC (58.48%) is significantly smaller than HC (67.54%) in the range of 0.1 V–3 V. Considering the raised surface area of PVP-HC, the sodium storage mechanism in our system is more in line with “adsorption-intercalation model” [39,40]. Larger specific surface area is expected to provide more adsorption capacity. Furthermore, impedance testing is confirmed to the charge transfer difference. As represented in Figure 5b, the R_ct_ (charge transfer resistance) of CTAB, PVP-HC are slower than the HC. Compounding with GO can accelerate ion transport and reduce polarization in electrode materials. This is also one of the key factors to improve the electrochemistry of the whole system. Multi-factor enhancements lead to improved cycle and multiplier performance.

## 3. Materials and Methods

*Materials preparation:* The tannin extract was directly pretreated at 500 °C (argon atmosphere) for one hour (heating rate: 5 °C per minute), then ground to a powder and sieved. The treatment was further carried out at 1000 °C for one hour to obtain the final product named HC. The product (1 g) and graphene oxide (0.2 g) were mixed in the configured solvent (100 mL of distilled water and 20 mL of ethanol), and then 0.05 g of PVP was added and stirred for 5 h. The material obtained was then dried under vacuum and treated in a tube furnace(Kejing, Hefei, China) at 1000 °C for one hour to obtain the final product named CTAB-HC. The PVP was replaced by CTAB with the same dosage and procedure, and the product obtained was named PVP-HC.

*Material characterization:* The morphology and structure of HC, CTAB-HC and PVP-HC were tested by scanning electron microscope (SEM JEOL JEM 2100F, Tokyo, Japan) at. The specific surface area of the samples was confirmed by ASAP 2460(Micromeritics, America) through nitrogen physisorption. Thermogravimetric analysis was analyzed by TG 209(Netzsch, Saarbrücken, Germany) under the atmosphere of N_2_ with a speed rate of 10 °C min^−1^. The crystal structure was recorded by Ultima IV (Cu-Kα radiation, Rigaku, Japan) under the rate of 10 °C min^−1^.

*Electrochemical analysis*: The electrochemical tests were all based on coin cells of type 2016, using lithium metal as the counter electrode. The preparation process of the working electrode includes: active material (name), conductive agent, and binder are mixed well in a solvent in the mass ratio of 70:15:15. The slurry was applied to the copper foil and vacuum dried at 120 degrees for 10 h. The electrolyte used was 1M LiPF6 in ethylene carbonate, diethyl carbonate, and dimethyl carbonate (1:1:1 by volume ratio). The electrochemical cycling and rate performance tests were performed in a land instrument(Land, Wuhan, China), and CV and EIS tests were done in an Ivium instrument(Ivium, Netherlands).

## 4. Conclusions

In summary, tannin was calcined to obtain HC within this work, and the results showed that it was irregular particles with little micro/mesopores on the surface, resulting in a small specific surface area. When HC was applied as the anode in LIBs, the coulombic efficiency was up to more than 83.68%, and the capacity can reach 218.1 mAh g^−1^ after 200 cycles (current density = 400 mA g^−1^). Further, compounding with graphene oxide, which is coated on the hard carbon surface, enhances the contact with the electrolyte and increases the lithium storage active site, while the kinetics are enhanced and the cycling performance and rate performance of the composite are improved. The lithium storage mechanism of hard carbon obtained in this system is more consistent with the adsorption-insertion mechanism. This article provides a facile method to prepare tannin-based hard carbon, which offers the possibility of low-cost and high-performance lithium anode in the future. Meanwhile, there is still much room to explore the electrochemical mechanism of tannin as hard carbon.

## Figures and Tables

**Figure 1 molecules-27-06994-f001:**
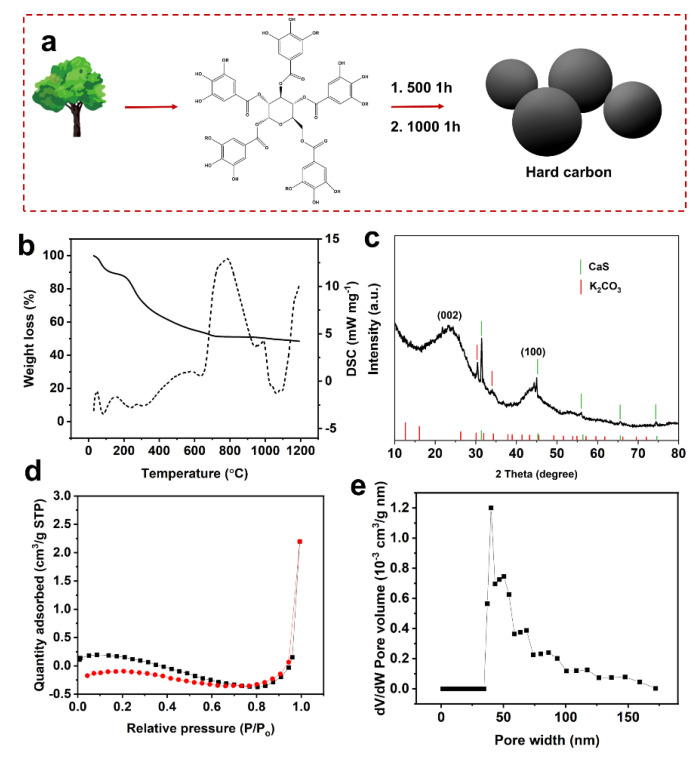
(**a**) Schematic diagram of the entire process for tannin to hard carbon. (**b**) Thermogravimetry and DTG curve of tannin. (**c**) XRD pattern of tannin-derived hard carbon. (**d**) N_2_ adsorption-desorption isotherms of tannin-derived hard carbon. (**e**) Pore size distribution curve of the HC.

**Figure 2 molecules-27-06994-f002:**
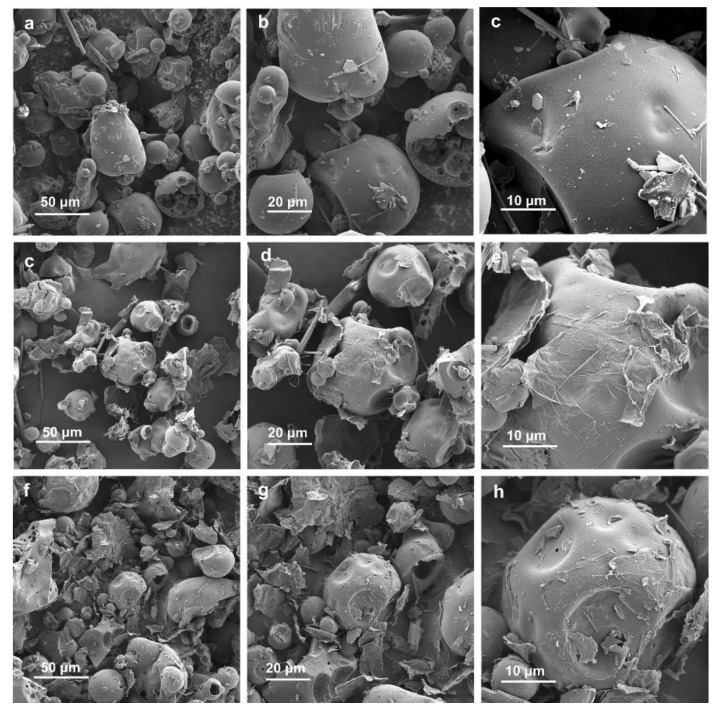
SEM images with different magnifications of (**a**–**c**) HC, (**c**–**e**) PVP-HC, (**f**–**h**) CTAB-HC.

**Figure 3 molecules-27-06994-f003:**
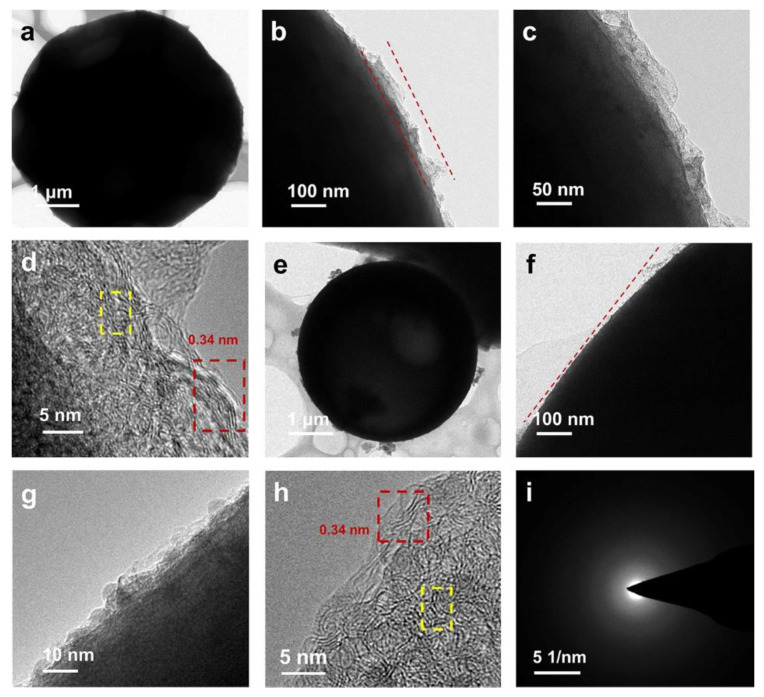
(**a**–**c**) TEM images of the CTAB-HC. (**d**) HRTEM image of the CTAB-HC. (**e**–**g**) TEM images of the PVP-HC. (**h**) HRTEM image of the PVP-HC. (**i**) SAED image of PVP-HC.

**Figure 4 molecules-27-06994-f004:**
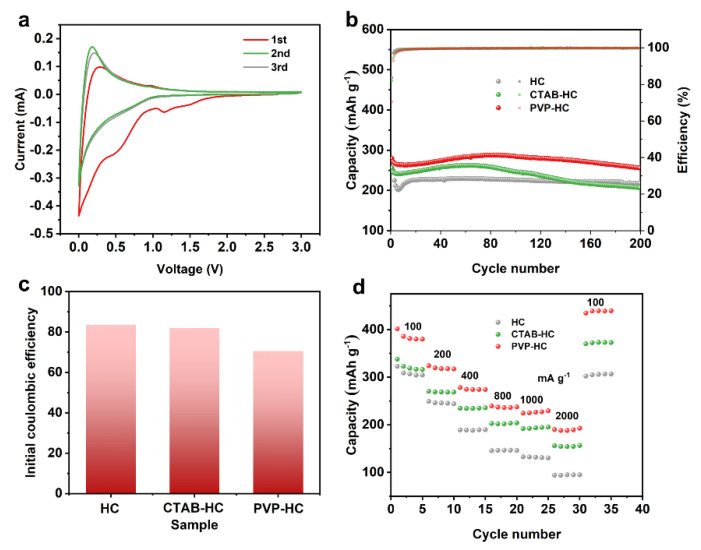
(**a**) Cyclic voltammetry curves of PVP-HC (0–3 V). (**b**) Cycling performance of HC, CTAB-HC, PVP-HC at the current density of 400 mA g^-1^. (**c**) Comparison of initial coulombic efficiency. (**d**) Rate performance of HC, CTAB-HC, PVP-HC at different densities.

**Figure 5 molecules-27-06994-f005:**
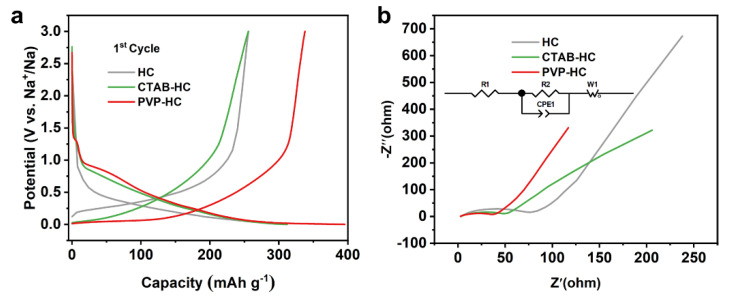
(**a**) Charge/discharge curves of the first cycle of HC, CTAB-HC, PVP-HC. (**b**) Nyquist plots of HC, CTAB-HC, PVP-HC.

## Data Availability

The data used to support these findings have been included in this article. Additional information is available from the corresponding authors upon request.

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
