# Peer review of "Tannin-Derived Hard Carbon for Stable Lithium-Ion Anode"

_molecules, 2022, doi:10.3390/molecules27206994_

Round 1

Reviewer 1 Report

In the present work, the authors reported the application of a Tannin-derived hard carbon for LiB applications. Overall, this work is well-organized and can be accepted for publication after minor modifications.

1.      Author used graphene oxide in the material preparation. How would the mass ratio between the HC and GO impact the structural properties and performance?

2.      XRD study revealed the existence of apparent impurities. What are their composition and contents? Where exactly are these impurities located? How would such impurities affect the electrochemical performance?

3.      The size of the carbon particles in SEM and TEM characterization doesn’t match with each other.

4.      At line 128, the author mentioned that "Furthermore, through the inverse fourier transform, the d002 is calculated to be 0.382 nm". From which data was the Fourier (F should be capitalized) transform performed?

Author Response

Reviewer: 1

Comments to the Author

Thank you for your useful comments and kind suggestions to improve our manuscript. We have revised the manuscript accordingly, and the detailed corrections are listed below point by point (All corrections have been noted in yellow in our revised manuscript):

In the present work, the authors reported the application of a Tannin-derived hard carbon for LiB applications. Overall, this work is well-organized and can be accepted for publication after minor modifications.

  1. Author used graphene oxide in the material preparation. How would the mass ratio between the HC and GO impact the structural properties and performance?

Our answer:

An appropriate amount of GO can coat hard carbon well and achieve performance improvement. Too much graphene will be used as the active material. Based on the interlayer spacing of graphene similar to graphite, it is not conducive to sodium storage, and the overall sodium storage performance will decrease.

  1. XRD study revealed the existence of apparent impurities. What are their composition and contents? Where exactly are these impurities located? How would such impurities affect the electrochemical performance?

Our answer:

According to the XRD patterns, it can be judged that the impurities are mainly composed of CaS and K2CO3. Inactive impurities will affect the capacity, but impurities may also affect the size of d002, and may also affect the degree of graphitization. There is uncertainty about the impact of battery performance, more thinking and exploration of this question is necessary in the future work.

  1. The size of the carbon particles in SEM and TEM characterization doesn’t match with each other.

Our answer:

Scanning electron microscope showed that the particle size was not uniform, and one of the particles was selected in transmission electron microscope to observe the coating structure.

  1. At line 128, the author mentioned that "Furthermore, through the inverse fourier transform, the d002 is calculated to be 0.382 nm". From which data was the Fourier (F should be capitalized) transform performed?

Our answer:

As can be seen from S3c-d in the figure, it is the average layer spacing obtained by inverse Fourier transform.

Reviewer 2 Report

In this work, the authors choose the tannin as the precursor to prepare hard carbon (HC). The as-prepared HC was investigated by the characterization technologies in depth. Graphene oxide was also introduced to further improve the electrochemical properties. This work expands the application of source-rich tannins as anode for lithium-ion batteries. The reviewer recommends it to be published after minor revision.

1. The XRD patterns of HC indicates the impurities on the carbon surface. What is the influence on the performance of HC?

2. The authors introduce GO to the system in two kinds of surfactant. Electrochemical properties also have corresponding differences. What is the source of the performance difference?

3. In Figure 5b, equivalent circuit diagram is required.

4. Some journals in the references have incorrect abbreviations. Such as “Journal of Physical Chemistry C”.

5. Some lithium-ion batteries-related work (Coordination Chemistry Reviews, 2020, 420, 213434; Electrochemical Energy Reviews, 2022, 5, 312-347.) could be cited to broaden readership.

Author Response

Reviewer: 2

Comments to the Author

In this work, the authors choose the tannin as the precursor to prepare hard carbon (HC). The as-prepared HC was investigated by the characterization technologies in depth. Graphene oxide was also introduced to further improve the electrochemical properties. This work expands the application of source-rich tannins as anode for lithium-ion batteries. The reviewer recommends it to be published after minor revision.

  1. The XRD patterns of HC indicates the impurities on the carbon surface. What is the influence on the performance of HC?

Our answer:

Good question! Inactive impurities will affect the capacity, but impurities may also affect the size of d002, and may also affect the degree of graphitization. There is uncertainty about the impact of battery performance, more thinking and exploration of this question is necessary in the future work.

  1. The authors introduce GO to the system in two kinds of surfactant. Electrochemical properties also have corresponding differences. What is the source of the performance difference?

Our answer:

In the PVP system, the graphene coating is more uniform and the effect is better. At the same time, the PVP system has a large specific surface area, which is conducive to the infiltration of the electrolyte and achieves better performance.

  1. In Figure 5b, equivalent circuit diagram is required.

Our answer:

The equivalent circuit diagram is added in Figure 5b.

  1. Some journals in the references have incorrect abbreviations. Such as “Journal of Physical Chemistry C”.

Our answer:

Thanks for comments! The related abbreviations have been modified in the right format.

  1. Some lithium-ion batteries-related work (Coordination Chemistry Reviews, 2020, 420, 213434; Electrochemical Energy Reviews, 2022, 5, 312-347.) could be cited to broaden readership.

Our answer:

As suggested, the relevant citations have been added in the revised manuscript.

Round 2

Reviewer 1 Report

This work can be accepted for publication now.